# Peppers under Siege: Revealing the Prevalence of Viruses and Discovery of a Novel Potyvirus Species in Venezuela

Eduardo Rodríguez-Román [1] , Yrvin León [1], Yearlys Perez [1] , Paola Amaya [1], Alexander Mejías [1],
Jose Orlando Montilla [2], Rafael Ortega [1], Karla Zambrano [3], Barlin Orlando Olivares [4],* and Edgloris Marys [1],*

1    Laboratorio de Biotecnología y Virología Vegetal, Centro de Microbiologia y Biologia Celular, Instituto
Venezolano de Investigaciones Científicas (IVIC), Caracas 1204, Venezuela;
erodriguezroman@gmail.com (E.R.-R.); yrvinln@gmail.com (Y.L.); yearlys@gmail.com (Y.P.);
amayapao2011@gmail.com (P.A.); alexandermejias@gmail.com (A.M.); rafaeskeo@gmail.com (R.O.)
2    Decanato de Agronomía, Cátedra de Fitopatología, Universidad Centroccidental Lisandro Alvarado (UCLA),
Barquisimeto 3001, Venezuela; jmontilla@ucla.edu.ve
3    Decanato de Agronomía, Cátedra de Biología, Universidad Centroccidental Lisandro Alvarado (UCLA),
Barquisimeto 3001, Venezuela; karlazambrano07@gmail.com
4    Biodiversity Management Research Group (GESBIO-UCO), Rabanales Campus,
University of Cordoba (UCO), Carretera Nacional IV, km 396, 14014 Cordoba, Spain
\*    Correspondence: ep2olcab@uco.es (B.O.O.); edgloris@gmail.com (E.M.)

**Abstract:** Many plant virus outbreaks have been recorded in the last two decades, threatening
food security around the world. During pepper production seasons in 2008, 2014, and 2022, virus
outbreaks were reported from Lara (western) and Miranda (central) states in Venezuela. Three
hundred seventy-three plants exhibiting virus-like symptoms were collected and tested for virus
infection through reverse transcription PCR (RT-PCR). The most prevalent viruses during the 2008
surveys conducted in Lara were potato virus Y (PVY, 66.25%), cucumber mosaic virus (CMV, 57.50%),
pepper mild mottle virus (PMMoV, 35%), alfalfa mosaic virus (AMV, 23.75%), and tobacco rattle
virus (TRV, 17.50%). This survey revealed for the first time that pepper is a natural host of AMV
and TRV in Venezuela. A further, divergent potyvirus isolate was also detected in 23.75% of pepper
plants from Lara state. In 2014, a follow-up survey after virus outbreaks reported in Lara and
Miranda states also detected this divergent potyvirus isolate in 21.68% of pepper plants, with tomato
spotted wilt virus (TSWV) and PMMoV dominating the viral landscape (62.65 and 21.68% of tested
plants, respectively). By comparison, the surveys revealed significant changes in viral community
composition. The complete capsid protein (CP) sequence of the putative potyvirus was obtained from
two pepper samples. According to the *Potyvirus* taxonomic criteria, these results suggest that the
isolate represents a distinct virus species, for which we propose the name "pepper severe mottle virus"
(PepSMoV). Virus outbreaks could be attributed to agricultural and environmental factors, such as
climate change, the use of wastewater, the use of uncertified seeds, misuse of agricultural chemicals,
transmission with food trade networks, and the development of new viral strains due to mutations
and recombination and pathogen spillover. This study demonstrates the value of knowledge of
the prevalence and distribution of viral species to recommend virus-resistant cultivars to replace
susceptible ones, especially in virus hotspot areas.

**Keywords:** diagnostic; disease; pepper; ribonucleic acid; survey; viruses





## 1. Introduction

Pepper (*Capsicum annuum* L.) belongs to the family *Solanaceae* and is an important
crop worldwide [1,2]. Pepper production in Venezuela for 2021 was 146,817 tons, with
a harvested area of 10,126 ha and a yield of 144,996 hg/ha [3]. The pepper crop faces
numerous challenges due to viral infections that can severely impact its production and
quality on a global scale [4,5]. In the central and western regions of Venezuela, pepper

cultivation serves as a significant agricultural activity, contributing to the nation's economy and food security. However, recent reports have raised concerns about the escalating prevalence of viral diseases in these areas, which is threatening the sustainability of pepper farming [6]. Hence, it is essential to identify the most commonly occurring and damaging viruses to recommend management strategies.

To date, 11 RNA viruses belonging to the genera *Cucumovirus*, *Ilarvirus*, *Nepovirus*, *Orthotospovirus*, *Potexvirus*, *Potyvirus*, *Tobamovirus*, and *Tobravirus* have been identified infecting pepper crops in Venezuela [6,7], mainly in the horticultural Lara state (located in the western region). Lara is the country's leading pepper producer where almost 70% of the production is achieved in the Quibor Valley. Multiple infections have been documented in the area, associated with an increased synergistic effect on the crop, leading to higher yield losses [7]. In 2008, viral outbreaks were reported by pepper producers in Quibor, where previously undescribed *Alfamovirus*, *Ilarvirus*, and *Potyvirus* species were detected.

During the spring of 2014, another major virus outbreak emerged in Quibor Valley (Lara) and in Altos Mirandinos (Miranda), the latter state now being considered the largest development pole of greenhouses dedicated to pepper production in Venezuela [8]. Pepper plants exhibiting virus-like symptoms were observed in different greenhouses throughout the states with incidences ranging from 21 to 62%. Remarkable pepper yield losses of up to 90% were associated with the emerging tomato spotted wilt virus (TSWV, *Orthotospovirus*) [6], causing field abandonment before harvest and making the cultivation of pepper not profitable. Preliminary observations by electron microscopy of crude sap from some of the diseased plants showed long flexuous filamentous particles of about 750 nm in length suggestive of viruses belonging to the genus *Potyvirus* (Appendix A, Figure A1). Moreover, the potyvirus "core" CP gene fragment obtained by RT-PCR from these plants exhibited 97–99% sequence identity with the above-mentioned isolates from 2008 surveys.

Given recurrent complaints from farmers facing heavy crop losses, molecular diagnostic assays were initiated to assess the relative importance of viruses infecting pepper in Lara and Miranda states. This paper aims to compare our findings during 2008, 2014 and 2022 pepper virus surveys in Lara and Miranda states. Also, in this study, we characterized the entire CP coding region of the newly isolated potyvirus. Phylogenetic analyses indicated that the virus was most closely related to pepper yellow mosaic virus (PepYMV). The results suggest that the virus should be classified as a novel species within the genus Potyvirus, which we tentatively name pepper severe mottle virus (PepSMoV). By shedding light on the viral landscape and uncovering this newly emerged pathogen, we aimed to enhance the understanding of the complex interactions between viruses and pepper plants, paving the way for effective management strategies and safeguarding the future of pepper cultivation.

## 2. Materials and Methods

### 2.1. Virus Surveys

Sample collection was carried out in Lara state in September 2008 and 2014 and in Miranda state in September 2014 and September 2022 (Table 1). Leaves and fruits showing virus-like symptoms (i.e., stunt, mottle, vein yellowing, mosaic, leaf distortion, light green leaves, necrotic spots on leaves or fruits, or distorted fruit showing brown streaks) were collected. Three or four apical leaves or fruits were collected from symptomatic crops. Immediately after collection, each sample was placed in a plastic bag, transported to the laboratory on ice, and stored at −80 °C or −20 °C pending analysis. Global positioning system (GPS) data were recorded at each greenhouse site, and a subsequent map was generated using QGIS v.3.28 [9].

**Table 1.** A number of pepper samples were collected from different regions of Venezuela.

| Sampling State | Year | Localities | Farms | Number of Samples ($n$) |
|---|---|---|---|---|
| Lara | 2008 | Tintorero | F10 to F13 | 80 |
| Lara | 2014 | Tintorero | F1 to F9 | 83 |
| Miranda | 2014 | Pozo de Rosas San Pedro El Jarillo | F1 to F6 | 108 |
| Miranda | 2022 | Pozo de Rosas, El Jarillo, Hoyo de La Puerta, La Reinosa | F7 to F10 | 92 |

*2.2. RNA Extraction, RT-PCR*

Total RNA was extracted from 100 mg of frozen plant tissue samples using the TRIzol^TM reagent (Life Technologies, Carlsbad, CA, USA) according to the manufacturer's recommendations. Approximately 1 µg of total RNA was used to generate first-strand cDNA with random, degenerated, or specific primers using a SuperScript III Reverse Transcription kit (Thermo Fisher Scientific Inc., Waltham, MA, USA) as reported [10–17] or designed in this study (Appendix A, Table A1). Each RT reaction contained about 200 ng of total RNA (2 µL), 1 µL of Random Oligo-dT (N6) primer or a specific reverse primer (10 µM), and 3 µL nuclease-free ddH$_2$O. RNAs from virus-infected samples used as positive controls were obtained from the Plant Virus Collection at IVIC. The PCR reaction (12.5 µL) contained 50 ng template cDNA, 10 pmol of each amplification primer, 200 mM each dNTP, 1.25U Taq DNA polymerase (Invitrogen, Carlsbad, CA, USA), 5 mM MgCl$_2$, and 10X Taq Polymerase PCR Buffer. Amplification parameters were as reported (Appendix A, Table A1). To determine the size of the amplified PCR products, a DNA ladder (Thermo Scientific, Waltham, MA, USA) was used. The RT-PCR products were examined by electrophoresis in 1% agarose gels containing ethidium bromide and examined and recorded using a Fotodyne UV/Digital camera transilluminator system.

*2.3. Cloning and DNA Sequencing*

Amplified PCR products were purified using the AccuPrep PCR Purification Kit (Bioneer, Seoul, Republic of Korea) and then cloned into the pGEM-T easy Vector System (Promega, Madison, WI, USA) according to standard methods [18]. Plasmid DNA preparations were obtained using the QIAprep Spin Miniprep Kit (Qiagen, Hilden, Germany). At least three clones representing a single PCR product were sequenced by Macrogen Inc. (Seoul, Republic of Korea). Sequences of the amplified PCR products were edited and assembled using MEGA7 [19]. The resulting sequences were used to BLAST search the sequence database at the National Center for Biotechnology Information (http://blast.ncbi.nlm.nih.gov/Blast.cgi, accessed on 5 March 2017).

*2.4. Molecular Characterization of Putative Potyvirus Species*

Some samples that tested positive only for the presence of a potyvirus with "core" CP primers showing 85% nt sequence identity to PepYMV were selected to amplify a larger portion of the genome, including the 3′ terminal sequence and the complete CP gene. The cDNA was synthesized using the primer B1570 Oligo(dT) complementary to the PepYMV polyadenylated tail. PCR was performed using the primer PY10, designed based on the pepper mottle virus (PepMoV) sequence, towards a conserved region in the nuclear inclusion body b (NIb) cistron, and B1570 [11] (Appendix A, Table A1). Expected fragments (ca. 1200 bp) comprising the CP and 3′ untranslated region (3′-UTR) of the potyvirus genome were cloned using the pGEM-T vector (Promega) according to the manufacturer's

recommendations. Two clones were selected and sequenced using M13F/M13R primers. The nucleotide (nt) and predicted amino acid (aa) sequences of the whole CP coding region and the nt sequence corresponding to the 3'-UTR were compared with potyvirus sequences deposited in GenBank, EMBL, DDBJ, and TrEMBL databases using the pairwise Align program. Sequence assembly and analysis were performed utilizing the DNA Dragon–Contig Sequence Assembly Software v1.7.1 [20]. Multiple sequence alignments produced by the Clustal W algorithm were used as input data for reconstructing phylogenetic trees by the neighbor-joining method using the software MEGA version 4 [21] (Table A2). Statistical significance was estimated by performing 1000 replications of bootstrap resampling of the original alignment using the bootstrap option of the phylogenetic tree menu.

### 2.5. Mechanical Transmission of Putative Potyvirus Isolate

To demonstrate the infectivity of the new potyvirus, frozen leaf tissues (0.3 g) from a single-infected potyvirus sample, AMPIM8, were used for mechanical inoculation of 10-day-old seedlings of *C. annuum* cv. Magistral. Sap was extracted from frozen samples by grinding tissue of leaves into a cold 0.1 M potassium phosphate buffer pH 7.0 containing 1% magnesium trisilicate, using a pre-chilled mortar. The homogenate was inoculated by gently rubbing the bottom leaves of healthy, carborundum-dusted pepper seedlings. Mock-inoculated plants were used as controls. Plants were subsequently grown in an insect-proof greenhouse until symptom expression. Four weeks after inoculation, virus infection was tested by RT-PCR using PepSMoV-specific primer pair PepSMoV-CP-F/PepSMoV-CP-R (Table A1) followed by Sanger sequencing. The virus from *C. annuum* cv. Magistral was reinoculated onto *C. annuum* cv. Magistral and cv. Acero for the fulfillment of Koch's postulates. Plants were tested by RT-PCR and Sanger sequencing for PepSMoV infection.

### 2.6. Analysis of Principal Coordinates (PCoA)

The analysis of principal coordinates was applied to assess the distribution of viruses across various farms (designated as F1 to F13) in the Miranda and Lara states of Venezuela through InfoStat software v.11 [22]. The methodology involved encoding the presence and absence of different viruses, including PMMoV, TSWV, PepSMoV, AMV, CMV, TRV, and PVY, as binary values (1 for presence and 0 for absence), using the distance matrix obtained from the S transformation $(1-S_{ij})^{1/2}$ with S = Dice's similarity index, to which a minimum spanning tree was superimposed to facilitate the visualization of the ordering. These binary values were then subjected to principal coordinate analysis, which aimed to uncover patterns and relationships in the viral distribution among the surveyed farms.

## 3. Results

### 3.1. Field Survey Results

Pepper fields were surveyed for virus diseases in Lara state in 2008 and 2014 and in Miranda state in 2014 and 2022 (Table 1, Figure 1). A total number of 373 pepper samples were collected during the surveys. The most common symptoms on the infected pepper plants during the 2008 surveys in Lara were stunting, mottling or mosaic, yellowing, and distortion in leaves (Figure 1b, left), whereas in 2014, the main symptoms found were ringspots on leaves and fruits and necrosis of leaves (Figure 1b, right). In Miranda state, chlorotic line patterns with necrotic spots were observed in mature plants, often showing cupped downward leaves (Figure 1d, left). Severe stunting of younger plants with chlorotic mosaic or yellow flecking of the leaves was also observed during 2022 surveys (Figure 1d, right). Based on symptomatology, an average virus incidence of 70–100% was recorded during surveys.

Surveys from Lara state in 2008 revealed the presence of five viruses; PVY was the most prevalent virus (66.25%), followed by CMV (57.50%), PMMoV (35%), AMV (23.75%), and TRV (17.50%). In this survey, AMV (GenBank accession OR420758) and TRV (GenBank accession OR420759) were found for the first time in pepper fields in Venezuela. The primer

pair MJ1–MJ2 used for potyvirus diagnosis also amplified a 327 bp fragment from 40 pepper samples (10.70%) that showed 85% nucleotide (nt) sequence identity to PepYMV.

Data from surveys in Lara state (2014) indicated that out of all samples showing virus-like symptoms, 75.9% (63/83) were positive for at least one virus. *Orthotospovirus* was the most common genus identified in 52 samples (62.6%), with TSWV being the only species found. *Tobamoviruses* were detected in 18 samples (21.6%), with PMMoV being the only species found. Potyvirus isolates sharing 85% nucleotide (nt) sequence identity to PepYMV, and 99–100% nt sequence identity to potyvirus isolates from 2008 in Lara, were also detected with the primer pair MJ1-MJ2 in 11 of the samples (13.2%), indicating relatively low prevalence. Single infections were more frequent than mixed infections (96.3 and 8.4%, respectively). TSWV and potyvirus coinfections were the most frequent mixed infections for all samples (4.8%), while triple infections (*Orthotospovirus–Tobamovirus–Potyvirus*) occurred at even lower percentages (3.6%). PMMoV and potyvirus coinfections were not recorded. A similar scenario occurred in Miranda in the same year, where TSWV had the highest overall prevalence (54.62%), followed by PMMoV (34.25%) and the divergent potyvirus species (23.14%). Double and triple infections were also recorded. These virus species (TSWV, PMMoV, and the divergent potyvirus) accounted for the viral population in Miranda during recent (2022) surveys.

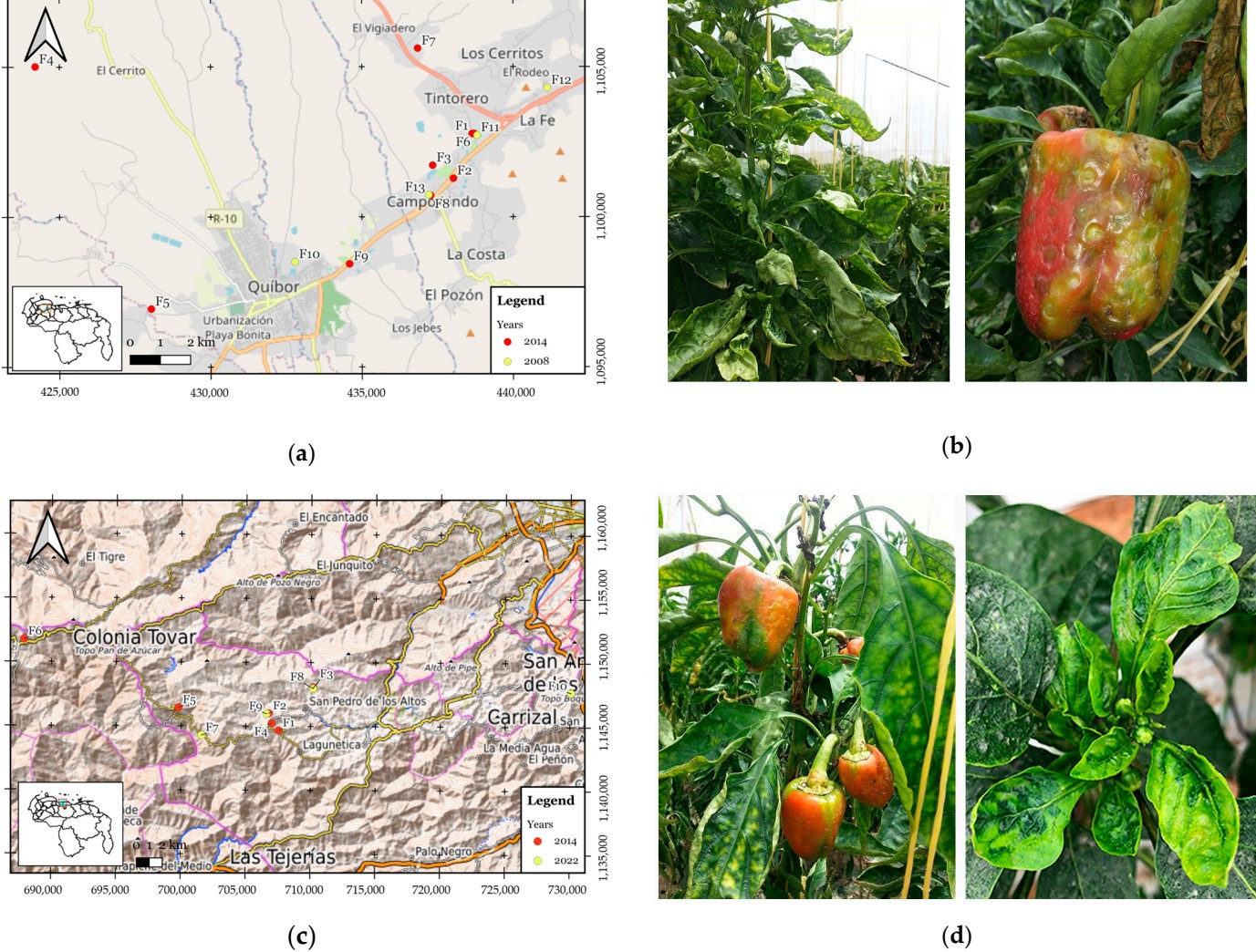

(**a**)

(**b**)

(**c**)

(**d**)

**Figure 1.** Virus detection using field-collected pepper samples in Lara Farms (F1 to F13) (**a**) and Miranda Farms (F1 to F10) (**c**), Venezuela. (**b**) Pepper surveys in Lara showed mosaic, yellowing of veinlets, and leaf deformation (**left**) during 2008 surveys and chlorotic spots on fruits and leaves (**right**)

during 2014 surveys. (**d**) Pepper surveys in Miranda show cupped downward leaves and chlorotic spots on fruits (2014, (**left**)) and severe mottling (2022, (**right**)). Pictures by Edgloris Marys (Miranda and Trujillo, Venezuela).

The circular dendrogram (Figure 2a) presents data on the distribution of various viruses in different regions and farms within the Miranda and Lara states of Venezuela. In Lara state, the virus population structure varied across different farms (F1 to F13). For instance, TSWV was detected in varying numbers in different farms, with the highest detection being 15 out of 28 samples in Miranda (F1). Similarly, PMMoV, PepSMoV, AMV, CMV, TRV, and PVY were also detected in different proportions across these farms. In Miranda state, a similar pattern emerged. PMMoV, TSWV, and PepSMoV were detected across different farms (F1 to F6), wth varying numbers of positive samples. In Miranda state (2022), farms F7 and F10, for example, appear to have higher virus prevalence compared to the others, indicating potential challenges for pepper plant health in those areas. In farm F7, for PMMoV, there were 11 positive samples out of 26 plant samples tested, and for TSWV, there were 13 positive samples out of 26 plant samples tested. In F10, for TSWV, there were 10 positive samples out of 20 plant samples tested. Overall, the analysis of virus population structures in different surveyed regions within Miranda and Lara states revealed fluctuations in the prevalence of various viruses across farms and variables.

The percentages provided for each virus reflect the proportion of pepper samples in which each virus was detected (Figure 2b). Notably, TSWV had the highest detection rate, being present in 41.6% of the samples, while PMMoV followed with a detection rate of 30.7%. PepS MoV was found in 20.8% of the samples, while PVY was detected in 19.6% of the samples.

### 3.2. Analysis of Principal Coordinates (PCoA)

Figure 3 displays the outcomes of the analysis of principal coordinates (PCoA) and the resulting tree of minimum distance, based on the presence (1) and absence (0) values of various viruses within different surveyed farms in Miranda and Lara states of Venezuela. The graph built with the first two axes of the PCoA explains 65.5% of the total variability in the presence or absence of the studied viruses. The analysis revealed that CP1 explained 39.4% of the variance, and farms F10, F11, F13 (Lara), and F9 (Miranda) had the highest scores along this coordinate, suggesting a common pattern of virus occurrence in these farms, while CP2 accounted for 26.1% of the variance, and farms F8 (Miranda) and F6 and F9 (Lara) had the highest scores along this coordinate, indicating a distinct pattern of virus distribution compared to other farms.

These principal coordinates offer insights into the relationships between the surveyed farms based on the virus presence/absence data. The tree of minimum distance illustrates the clustering of farms that have similar virus compositions, indicating potential patterns of virus distribution within these regions. The presence or absence of viruses within farms was used to create a distance metric, and the resulting tree visually represents the similarity between farms in terms of their virus populations.

### 3.3. Characterization of Potyvirus Isolates

The detection of potyvirus was carried out using genus-specific primers MJ1 and MJ2, designed to amplify a short 327 nt fragment spanning conserved motifs MVWCIEN to QMKAAA in the "core" of the CP of potyviruses. The primers were chosen because they gave superior amplification signals in preliminary experiments. Following the removal of primer sequences, the resulting 324 bp fragments obtained from 11 positive plants collected in Lara in 2014 (GenBank accessions MH785274 to MH785295) showed a unique 85% nucleotide (nt) sequence identity to PepYMV isolated from *Capsicum* sp. in Brazil (AF348610); 82% to pepper severe mosaic virus (AM181350.1); 81% to potato virus V (KP849483.1); 80% to Peru tomato mosaic virus (AJ516016.1), brugmansia mosaic virus (JX867236.1), and Amazon lily mosaic virus (AB158523.1); and 79% to pepper mottle

virus (EU586135.1), Ecuadorian rocoto virus (EU495234.1), mashua virus Y (MH680823.1), verbena virus Y (EU564817.1) and *Amaranthus leaf mottle virus* (AJ580095.1).

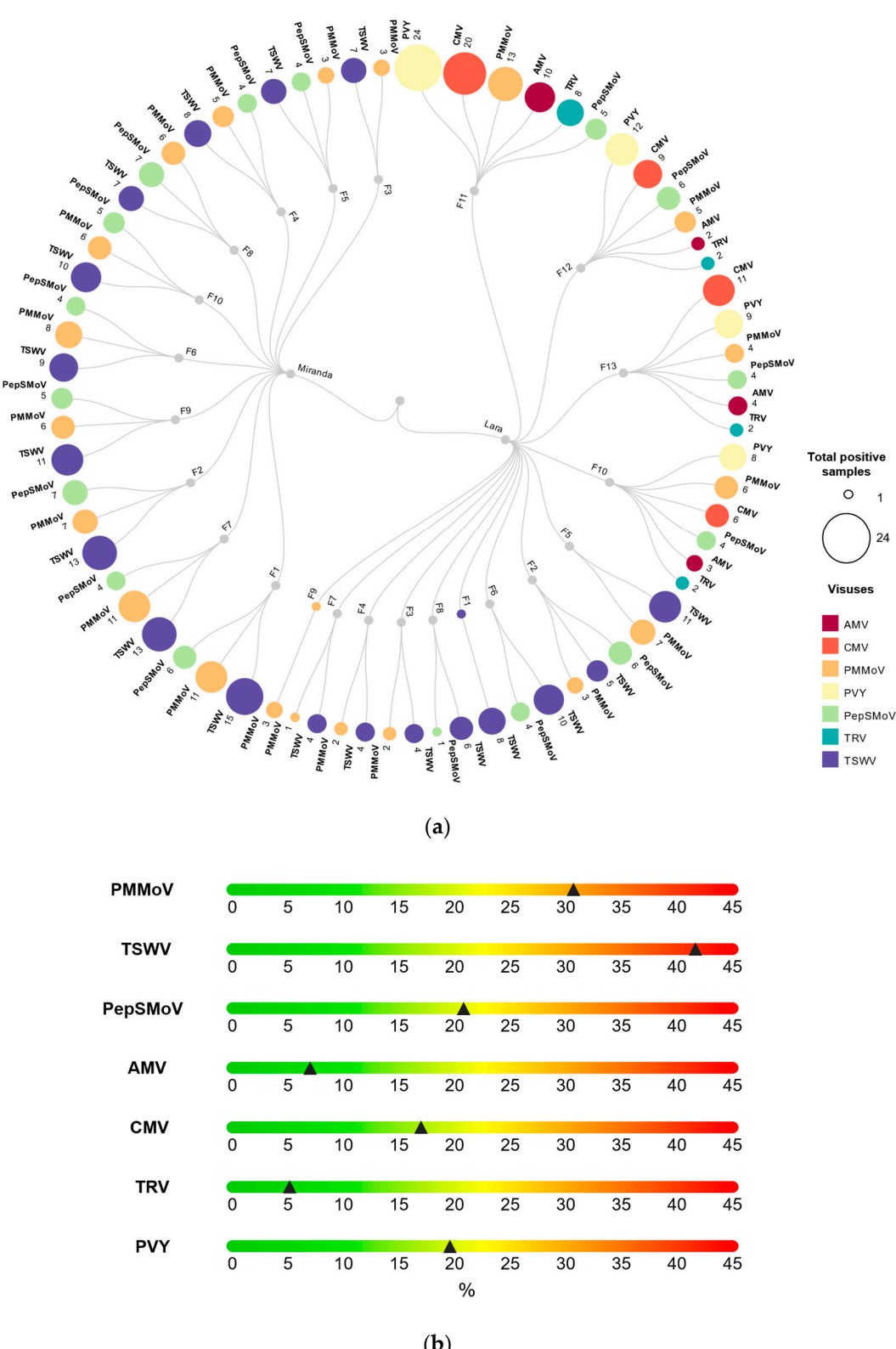

**Figure 2.** (**a**) Circular dendrogram of virus population structures in different regions and Farms (F1 to F13) surveyed in this study. (**b**) Detection rates of viruses (%) in pepper samples.

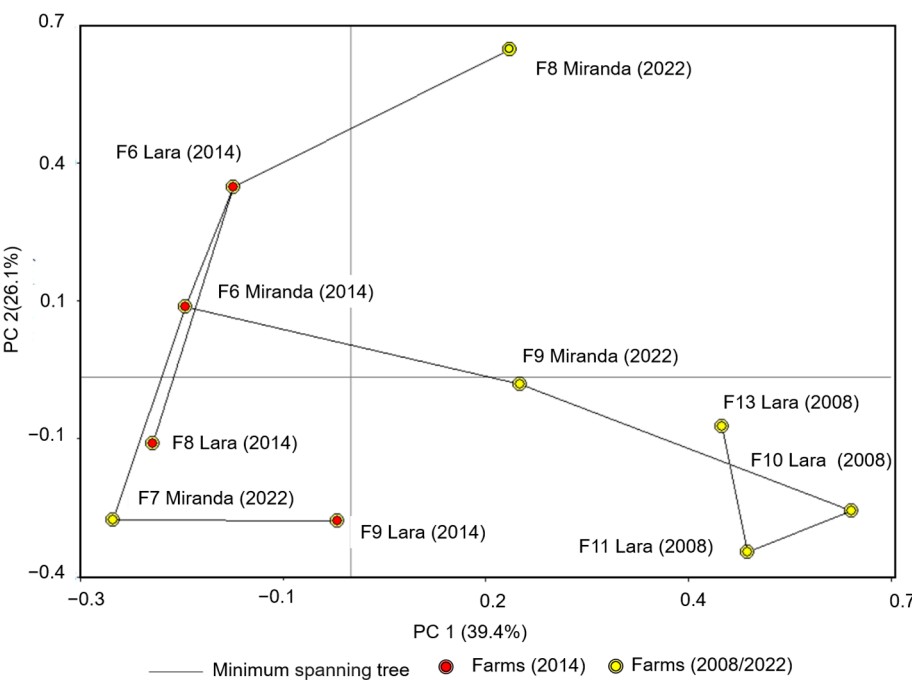

**Figure 3.** Scatter diagram from the main coordinates (PC1 and PC2) obtained using "Dice" distances between the farms based on the binary presence and absence of viruses. The minimum spanning tree is superimposed.

To further characterize the divergent potyvirus isolates, a larger RT-PCR fragment (ca. 1.163 bp) comprising the CP and 3′ untranslated region (3′-UTR) of the potyvirus genome was obtained and sequenced (KT721567.1) using primers designed for PepYMV from single-infected samples. Then, the 3′ terminal sequence was split into the NIb, CP, and 3′-UTR according to the inferred cleavage site (-VHHQ/AD-) and stop codon (TAA). The resulting 837 nt CP sequence was subjected to BLASTn and BLASTx analysis and indicated 73.9% and 76.7% nt and aa identity to PepYMV (NC_014327). These sequence identities met the current species demarcation criteria for the *Potyvirus* genus [23]. These findings, therefore, suggest that our isolate is a new potyvirus species that possesses a closer evolutionary relationship with potyvirus PepYMV. The CP is 277 aa long, and analyses showed that the well-known -DAG- motif, which is involved in aphid transmission, was absent from the CP sequence. Instead, a -DAA- motif, which is also present in PepYMV-CP [24], was identified. The deduced amino acid sequence of the CP was most similar to the CP of some potyviruses within the potyvirus supergroup (Table 2).

**Table 2.** Identity and similarity (%) of the deduced amino acid sequence of the divergent potyvirus to those of other potyviruses.

| Virus/Genbank Accession | Identity | Similarity |
|---|---|---|
| PepYMV (NC_008393.1) | 76.7 | 83.0 |
| EcRV (EU495234.1) | 71.0 | 82.4 |
| PeSMV (NC_008393.1) | 74.8 | 82.3 |
| PTV (NC_004573.1) | 73.1 | 83.5 |
| PepMoV (NC_001517.1) | 69.9 | 79.9 |
| PVY (NC_001616.1) | 73.5 | 81.0 |
| TEV (NC_001555.1) | 58.6 | 75.4 |
| ChiRSV (NC_016044.1) | 57.0 | 74.9 |
| ChiVMV (NC_005778.1) | 57.7 | 72.9 |
| PVMV (NC_011918.1) | 58.4 | 76.3 |
| WTMV (NC_009744.1) | 59.4 | 74.1 |

To further investigate the evolutionary relationship between our isolates and other potyviruses, we constructed phylogenetic trees at the CP protein level. According to the phylogenetic tree, our isolates were placed in a separate branch closer to PepYMV isolates within the potyvirus supergroup (Figure 4). These findings, therefore, suggest that our isolate represents a new species of *Potyvirus* that possesses a closer evolutionary relationship with PepYMV. We suggest the name pepper severe mottle virus (PepSMoV) for the novel isolate.

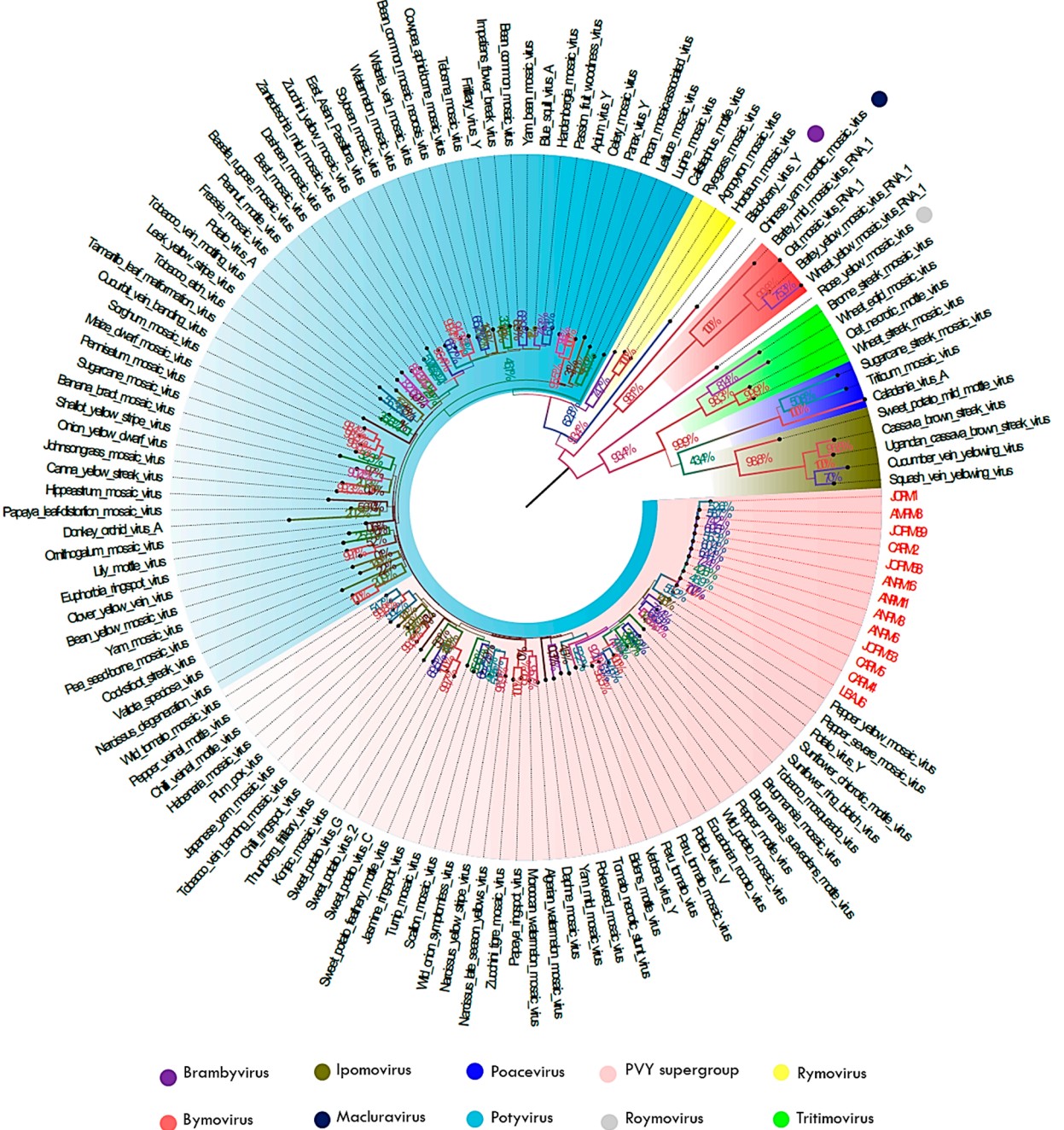

**Figure 4.** Phylogenetic tree illustrating the position of novel potyvirus isolates from pepper (in red) among the members of the family *Potyviridae*. The tree was constructed from multiple sequence alignments of capsid protein amino acid complete sequences obtained from Genbank, using the LG + G model. Bootstrap values (*n* = 1000) or probability estimate values larger than 70% are indicated at branch nodes for neighbor-joining/maximum-likelihood/Bayesian analysis. Each colored box indicates major phylogenetic groups.

In single virus infections, the new potyvirus isolates induced leaf distortion, severe mottling, and mosaic symptoms on systemic leaves of *C. annuum* cv. Magistral 14 days post-inoculation (dpi) (Figure 5a). The symptoms resemble those observed in single-infected PepSMoV found in pepper fields (Figure 5b). Single RT-PCR analysis of mechanically inoculated plants confirmed potyvirus infection. To fulfill Koch's postulates, a virus from *C. annuum* cv. Magistral was reinoculated onto *C. annuum* cv. Magistral and cv. Acero. Severe mottling was observed in new leaves at 14 dpi, which matched the symptoms found in the field. Reinoculated host plants were positive by RT-PCR only when tested with specific PepSMoV primers (Figure 5c), indicating that the mottling symptoms were caused by only one kind of plant virus.

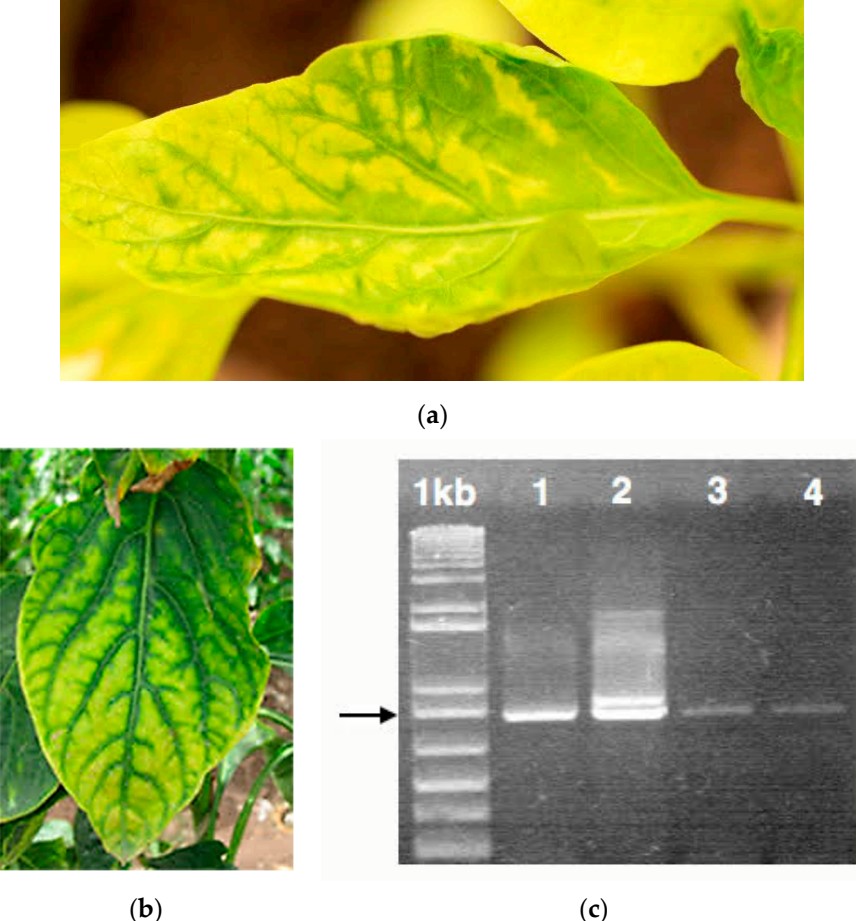

(**a**)

(**b**)　　　　　　　　　　　　　　　(**c**)

**Figure 5.** (**a**) Pepper plants (*Capsicum annuum* L.) var. Magistral mechanically inoculated with PepSMoV showing severe mottling and yellowing on leaves. (**b**) Single-infected PepSMoV in the field. (**c**) Amplification of 837 bp PepSMoV-CP fragments from the field-infected sample AMPIM8 (1); inoculated *C. annunm* var. Magistral (2); reinoculated *C. annunm* var. Acero (3, 4). The arrow points to the 850 bp molecular weight marker. Pictures by Edgloris Marys (Miranda and Trujillo, Venezuela).

## 4. Discussion

Although quantification of crop losses due to pathogens in Venezuela is limited, plant disease outbreaks are causing substantial declines in major staple food and cash crops, and this impacts rural livelihoods and poses a significant and growing threat to the already complex food insecurity crisis in the country. This work highlights the importance of virus detection in pepper—one of the most popular species of vegetables in Venezuela—in Lara and Miranda states, which account for most of Venezuela's pepper production. Epidemiological knowledge on viral diseases of pepper has been accumulated poorly in Venezuela during the last few decades. Previously, the use of ELISA assays had limited our knowledge of viruses infecting pepper in the country and did not generate molecular

evidence to confirm the occurrence of specific viruses [7]. In this study, most samples were infected with at least one virus (76.0%), while 24.0% of the samples remained negative. Symptoms observed in negative samples could be related to certain physiological and nutritional disorders, phytotoxicity or senescence, or possibly to other viruses not tested for. A previous survey on the presence and incidence of pepper viruses in fields located in Lara state carried out many years ago [7] showed that PMMoV and TEV were the most prevalent viruses (100%), followed by ToRSV (58.2%), TRSV (50.0%), CMV (35.29%), TMV (26.47%), PVY (17.65%), and TRS (14.72%). Thus, it was reasonable to focus on determining their incidence in this research. Surveys from 2008 revealed the presence of AMV and TRV, which constitutes—to our knowledge—the first report of this virus infecting pepper crops in Venezuela, which could pose a threat to other economic crops grown in Lara. However, AMV and TRV were not detected in the 2014 surveys, demonstrating the importance of disease surveillance and monitoring systems.

Data obtained in this studio reveals a shift in virus prevalence with the emergence of TSWV, only recently reported in the country [6], with incidence rates as high as 62.3% which could explain the nature and severity of the symptoms reported by farmers. TSWV is considered a major agricultural pest because of its worldwide distribution, wide host range, significant crop losses resulting from infection, and the difficulty in managing the thrip vector [25]. The factors driving the emergence and spread of TSWV in Lara and Miranda remain largely unknown but could be attributed in part to high thrip (*Frankliniella occidentalis*) densities recorded during sampling.

Interestingly, PMMoV remains one of the most common viruses detected in pepper. A decrease in PMMoV incidence from 2004 (100%) to 2014 (21.6%) could be attributed to the fact that—at least until 2015—a considerable number of farmers changed the practice of using their seed to buying the seed of modern hybrid cultivars. This confirmed the importance of seed origin, quality, and health status in the control of seed-transmitted *Tobamovirus* [26]. TEV (genus *Potyvirus*), whose presence in pepper fields in Jimenez has been previously established in Venezuela [7], was not detected in this work. In contrast with the 2004 survey, symptomatic samples in this study tested negative for ToRSV, TRSV, CMV, TMV, and TRS. Data on pepper variety were not collected during surveys; however, it is known that "Majestic", a highly TMV-resistant pepper variety, was the most popular variety being grown by farmers in previous years. Although there is a lack of information on the season when the 2004 survey took place, this is likely due to the different seasons in which samples were collected. In 2014, the surveys were conducted in September (end of the rainy season; moderate temperatures). As suggested by Afouda et al. [27], the abundance of virus vector populations (aphids in the case of CMV and PVY) likely correlates with seasonal variations affecting virus incidence. Nematode-transmitted virus (ToRSV, TRSV, TRS) incidence also could be influenced by seasonal fluctuations in the spatial distribution of the nematode population [28] and by the change in the pepper production system.

The varying detection rates of these viruses in the pepper samples from Miranda and Lara states can also be attributed to a combination of factors. The geographic distribution and environmental conditions of the two states may have influenced the prevalence of these viruses. Different viruses could have been favored by specific climatic conditions [29,30], temperature ranges [31], and soil types present in these regions [32]. Additionally, the population dynamics of insect vectors, such as aphids and thrips, may have contributed to the differing transmission rates of these viruses. The presence of abundant vector populations could lead to higher rates of virus transmission [33]. Furthermore, variations in host plant susceptibility might have played a role. Different pepper cultivars could have exhibited varying levels of resistance to certain viruses, leading to differences in infection rates. The movement of infected plant material, such as seeds and seedlings, could also have contributed to the spread of viruses across the two states. Additionally, cultural practices and farming strategies employed in each region could have influenced the transmission dynamics of these viruses. Interactions between viruses and possible coinfections within pepper plants might have affected detection rates as well. Some viruses could interact

synergistically or competitively, influencing their ability to establish infections within the same host plant [34].

A new potyvirus was found associated with the 2014 virus outbreak. Electron microscopy analysis of infected tissues showed flexuous filamentous particles in the 800–900 nm size range typical of a potyvirus. RT-PCR using *Potyvirus* genus-specific primers and subsequent sequencing identified the infectious agent as a potentially new virus species, PepSMoV, belonging to the genus *Potyvirus*. Further coat protein sequence analysis of PepSMoV and phylogenetic analysis with other viruses confirmed that PepSMoV belongs to the genus *Potyvirus*. The molecular criteria for species discrimination within the *Potyvirus* genus have been established by the International Committee on Taxonomy of Viruses (ICTV) [24]. The species demarcation criteria, based upon the large ORF or its protein product, are generally accepted as <76% nucleotide identity and <82% amino acid identity. Pairwise homology studies of CP genes were undertaken between PepSMoV and its closest related potyvirus, PepYMV. PepSMoV has a nucleotide identity of 74% and amino acid identity of 77% with PepYMV and meets the molecular species demarcation criteria.

Based on these criteria and the results obtained from BLAST and multiple alignments of nucleotide and amino acid sequences, with high certainty, we suggest the presence of a virus that belongs to a new species of Potyvirus that is most closely related to PepYMV, with which it shares 73.9% and 76.7% nucleotide and amino acid identity in the CP.

It is important to try to reveal the origin and evolutionary history of each virus. Like its closest relative, PepSMoV did not carry the DAG motif present in most potyvirus CPs as an important factor related to aphid transmission [35], both sharing instead a DAA motif [36]. This confirms that PepYMV and PepSMV are highly similar and could have a common origin.

PepYMV is a species indigenous and confined to Brazil, and it is the prevalent virus in pepper fields [37]. It is believed that the extensive use of PVY-resistant cultivars may have contributed to the emergence of PepYMV in that country [24].

Comparing the relatively low incidence of PepSMoV in the survey samples, we speculate that it might be a minor virus of pepper. Nevertheless, to determine its significance for the genus *Capsicum*, future diagnostic surveys in Venezuela should include testing for the presence of this virus. Issues such as complete genome sequence, host range, vector transmission [38], epidemiology [39], and pathological properties [40], relevant to the proper management of viral diseases in peppers, should also be addressed. This work constitutes the first attempt to determine the role of RNA viruses in pepper production in Lara state during the 2014 epidemic. It is expected that surveillance programs aimed at diagnosing and preventing virus spread will be implemented in our country.

## 5. Conclusions and Prospects

Despite yield losses, very few viral emergencies or novel threats have been mentioned in pepper crops in Venezuela during the past 20 years. Our study aimed to enhance the understanding of RNA virus prevalence in pepper plants, paving the way for effective management strategies and safeguarding the future of pepper cultivation. Our key findings include the following:

(1) Five virus species (PVY, CMV, PMMoV, AMV, and TRV) were identified in samples collected during the 2008 outbreak in Lara state. Two of these species (AMV and TRV) were found infecting pepper for the first time in the country. Synergistic disease caused by mixed virus infection could account for the crop losses reported during 2008.

(2) An alarming prevalence of TSWV and PMMoV in the surveyed regions during the 2014 and 2022 virus outbreaks reaffirms the severity of their impact on pepper production.

(3) Surveys revealed the unexpected and groundbreaking discovery of an entirely new potyvirus species previously uncharacterized. This previously unknown virus represents a significant addition to the existing repertoire of viral threats to pepper crops in Venezuela and potentially beyond.

The study made important contributions to sustainability in various dimensions. The research had a social impact by addressing viral threats to pepper crops, directly benefiting the agricultural communities of Lara and Miranda in Venezuela. The study helped us understand the threats posed by these pathogens. This contributes to sustainable agriculture by safeguarding crop health and minimizing yield losses due to viral infections. The study addressed a real-world agricultural challenge by revealing the prevalence of viruses in pepper plants. It provided essential information about the existing situation and the temporal evolution (2008, 2014, and 2022), which allowed the identification and quantification of viral threats.

The research benefited local farmers and agricultural stakeholders by offering for the first time (as far as is known) the discovery of a new species of potyvirus in Venezuela. It provided them with knowledge and certainty in the identification of the pathogen to protect their crops effectively. The discovery of a new species of Potyvirus expanded the theoretical knowledge base in virology and phytopathology. It contributed to the understanding of viral diversity and evolution, supporting and enriching theoretical foundations in these fields.

The study introduced new approaches to studying and managing viral infections in crops. It offered innovative methods for detecting viruses, which can be applied in research and agricultural practices beyond Venezuela. By presenting new approaches and solutions to address the prevalence of the virus, the research justified its practical importance. From this study, practical recommendations can be established that could be implemented on farms, minimizing losses and promoting sustainable agricultural practices.

The research findings and methodologies have the potential to be applied in other institutions, communities, or organizations beyond their original context. Their insights into the prevalence and management of the virus could be adapted to benefit other regions facing similar agricultural challenges, thus extending the positive impact and sustainability beyond Venezuela.

**Author Contributions:** Conceptualization, E.M., B.O.O. and E.R.-R.; methodology, E.M., E.R.-R. and B.O.O.; software, E.R.-R., Y.L., K.Z. and B.O.O.; validation, E.M., E.R.-R., Y.L. and B.O.O.; formal analysis, E.M., E.R.-R., Y.L. and B.O.O.; investigation, J.O.M., A.M., Y.P., Y.L., E.R.-R., P.A., E.M., E.R.-R. and R.O.; resources, E.R.-R., E.M. and B.O.O.; data curation, E.M., E.R.-R., Y.L. and B.O.O.; writing—original draft, E.M.; writing—review and editing, B.O.O. and E.R.-R.; visualization, E.R.-R. and B.O.O.; supervision, E.M.; project administration, E.M.; funding acquisition, E.M. All authors have read and agreed to the published version of the manuscript.

**Funding:** This study was supported by grants from the Inter-American Development Bank (IDB), the National Fund for Science and Technology (FONACIT, Venezuela), and the Venezuelan Institute for Scientific Research (IVIC) to project CMBC-408.

**Institutional Review Board Statement:** Not applicable.

**Informed Consent Statement:** Not applicable.

**Data Availability Statement:** Not applicable.

**Acknowledgments:** The authors are grateful to all the growers for field assistance.

**Conflicts of Interest:** The authors declare no conflict of interest.

## Appendix A

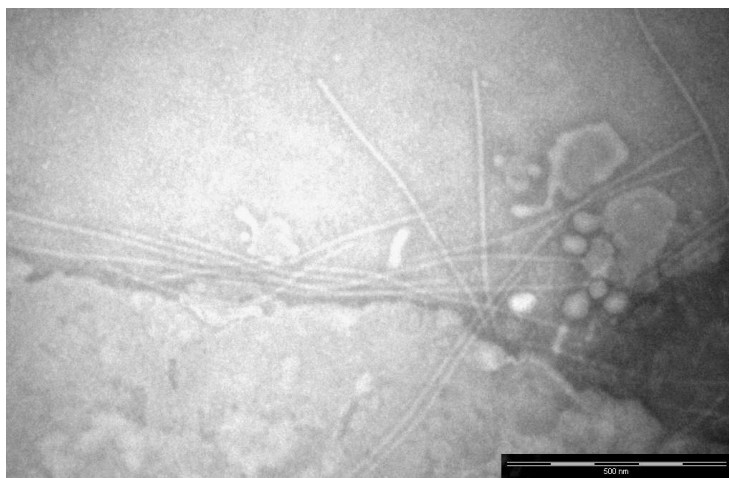

**Figure A1.** Electron micrograph of partially purified virions negatively stained.

**Table A1.** The list of primers used for the detection of pepper viruses.

| Virus Genus | Primer Name | Sequence (5′-3 * | Amplicon Size (bp) | Reference |
|---|---|---|---|---|
| *Ilarvirus* | Ilar1F5/ | GCNGGWTGYGGDAARWCNAC | 300 | [16] |
| | Ilar1R7 | AMDGGWAYYTGYTYNGTRTCACC | | |
| *Nepovirus* | Nepo-AF | GGHDTBCAKTMYSARRARTGG | 255 | [12] |
| | Nepo-AR | TGDCCASWVARYTCYCCATA | | |
| | Nepo-CF | TTRKDYTGGYKAAMYYCCA | 640 | [12] |
| | Nepo0CR | TMATCSWASCRHGTGSKKGCCA | | |
| *Orthotospovirus* | L1/ | AATTGCCTTGCAACCAATTC | 276 | [10] |
| | L2 | ATCAGTCGAAATGGTCGGCA | | |
| *Potexvirus* | Potex 1RC/ | TCAGTRTTDGCRTCRAARGT | 584 | [17] |
| | Potex 5 | CAYCARCARGCMAARGAYGA | | |
| *Potyvirus* | MJ1/ | TGGTHTGGTGYATHGARAAYGG | 327 | [14] |
| | MJ2 | TGCTGCKGCYTTCATYTG | | |
| | B1570/ | GGAGAGTCTTGGGCT | 1.200 | [11] |
| | PY10 | GCAATGCTTGAGTCATGGGG | | |
| | PepSMoV-CP-F | GCAGATGACACAAGTAAAACT | 837 | This study |
| | PepSMoV-CP-R | CATATTCTTCACCCCAAGCAA | | |
| *Tobamovirus* | TobUni1/ | ATTTAAGTGGASGGAAAAVCAT | 750 | [15] |
| | TobUni2 | GTYGTTGATGAGTTCRTGGA | | |
| *Tobravirus* | Tobra-F3/ | GGTGGKCAATGGTCTTWTTGG | 800 | [13] |
| | Tobra-R2 | GTCAGCTGYTGATCAGATAACC | | |

* R = A or G; Y = C or T; S = G or C; W = A or T; K = G or T; M = A or C; B = C or G or T; D = A or G or T; H = A or C or T; V = A or C or G; N = any base.

**Table A2.** Reference sequences used for phylogenetic analysis.

| Virus/Acronym | Genbank Accession No. |
|---|---|
| agropyron mosaic virus | NC_005903.1 |
| algerian watermelon mosaic virus | NC_010736.1 |
| apium virus Y | NC_014905.1 |
| arracacha mottle virus | NC_018176.1 |
| artichoke latent virus isolate FR37 | NC_026759.1 |
| asparagus virus 1 isolate DSMZ PV-0954 | NC_025821.1 |
| banana bract mosaic virus | NC_009745.1 |

**Table A2.** *Cont.*

| Virus/Acronym | Genbank Accession No. |
|---|---|
| barley mild mosaic virus RNA 1 | NC_003483.1 |
| barley mild mosaic virus RNA2 | NC_003482.1 |
| barley yellow mosaic virus RNA 1 | NC_002990.1 |
| barley yellow mosaic virus RNA 2 | NC_002991.1 |
| basella rugose mosaic virus | NC_009741.1 |
| bean common mosaic necrosis virus | NC_004047.1 |
| bean common mosaic virus | NC_003397.1 |
| bean yellow mosaic virus | NC_003492.1 |
| beet mosaic virus | NC_005304.1 |
| bidens mosaic virus isolate SP01 | NC_023014.1 |
| bidens mottle virus | NC_014325.1 |
| blackberry virus Y | NC_008558.1 |
| blue squill virus A | NC_019415.1 |
| brazilian weed virus Y isolate KLL097 | NC_030847.1 |
| brome streak mosaic virus | NC_003501.1 |
| brugmansia mosaic virus strain SK | NC_020105.1 |
| brugmansia suaveolens mottle virus | NC_014536.1 |
| caladenia virus A | NC_018572.1 |
| calla lily latent virus strain m19 polyprotein gene | NC_021196.1 |
| callistephus mottle virus isolate DJ | NC_030794.1 |
| canna yellow streak virus | NC_013261.1 |
| carrot thin leaf virus isolate CTLV-Cs | NC_025254.1 |
| cassava brown streak virus | NC_012698.2 |
| catharanthus mosaic virus isolate Mandevilla-US | NC_027210.1 |
| celery mosaic virus | NC_015393.1 |
| chilli ringspot virus | NC_016044.1 |
| chilli veinal mottle virus | NC_005778.1 |
| chinese yam necrotic mosaic virus | NC_018455.1 |
| clover yellow vein virus | NC_003536.1 |
| coccinia mottle virus isolate Su12-25 | NC_030840.1 |
| cocksfoot streak virus | NC_003742.1 |
| colombian datura virus | NC_020072.1 |
| cowpea aphid-borne mosaic virus | NC_004013.1 |
| cucumber vein yellowing virus | NC_006941.1 |
| cucurbit vein banding virus isolate 3.1 | NC_035134.1 |
| daphne mosaic virus | NC_008028.1 |
| dasheen mosaic virus | NC_003537.1 |
| donkey orchid virus A isolate SW3.1 polyprotein gene | NC_021197.1 |
| east asian passiflora virus | NC_007728.1 |

**Table A2.** *Cont.*

| Virus/Acronym | Genbank Accession No. |
|---|---|
| endive necrotic mosaic virus strain ENMV-FR | NC_034273.1 |
| ecuadorian rocoto virus isolate Rocoto | EU495234.1 |
| euphorbia ringspot virus isolate PV-0902 | NC_031339.1 |
| freesia mosaic virus | NC_014064.1 |
| fritillary virus Y | NC_010954.1 |
| habenaria mosaic virus genomic RNA | NC_021786.1 |
| hardenbergia mosaic virus | NC_015394.2 |
| hippeastrum mosaic virus | NC_017967.1 |
| hordeum mosaic virus | NC_005904.1 |
| hubei poty-like virus 1 strain SCM51506 polyprotein gene | NC_032912.1 |
| impatiens flower break potyvirus isolate Asan | NC_030236.1 |
| iranian johnsongrass mosaic virus | NC_018833.1 |
| iris severe mosaic virus isolate BJ | NC_029076.1 |
| japanese yam mosaic virus | NC_000947.1 |
| jasmine ringspot virus | NC_029051.1 |
| johnsongrass mosaic virus | NC_003606.1 |
| keunjorong mosaic virus isolate Cheongwon | NC_016159.1 |
| konjac mosaic virus | NC_007913.1 |
| leek yellow stripe virus | NC_004011.1 |
| lettuce italian necrotic virus | NC_027706.1 |
| lettuce mosaic virus | NC_003605.1 |
| lily mottle virus | NC_005288.1 |
| longan witches broom-associated virus isolate Han1 | NC_034835.1 |
| lupine mosaic virus | NC_014898.1 |
| maize dwarf mosaic virus | NC_003377.1 |
| moroccan watermelon mosaic virus | NC_009995.1 |
| narcissus degeneration virus | NC_008824.1 |
| narcissus late season yellows virus isolate Marijiiup8 | NC_023628.1 |
| narcissus yellow stripe virus | NC_011541.1 |
| oat mosaic virus RNA 1 | NC_004016.1 |
| oat mosaic virus RNA 2 | NC_004017.1 |
| oat necrotic mottle virus | NC_005136.1 |
| onion yellow dwarf virus | NC_005029.1 |
| ornithogalum mosaic virus | NC_019409.1 |
| panax virus Y | NC_014252.1 |
| papaya leaf distortion mosaic virus | NC_005028.1 |
| papaya ringspot virus | NC_001785.1 |
| passion fruit woodiness virus | NC_014790.2 |

**Table A2.** *Cont.*

| Virus/Acronym | Genbank Accession No. |
|---|---|
| pea seed-borne mosaic virus | NC_001671.1 |
| peanut mottle virus | NC_002600.1 |
| pecan mosaic-associated virus isolate LA | NC_030293.1 |
| pennisetum mosaic virus | NC_007147.1 |
| pepper mottle virus | NC_001517.1 |
| pepper severe mosaic virus | NC_008393.1 |
| pepper veinal mottle virus | NC_011918.1 |
| pepper yellow mosaic virus | NC_014327.1 |
| peru tomato mosaic virus | NC_004573.1 |
| plum pox virus | NC_001445.1 |
| pokeweed mosaic virus isolate PkMV-PA | NC_018872.2 |
| potato virus A | NC_004039.1 |
| potato virus V | NC_004010.1 |
| potato virus Y | NC_001616.1 |
| rice necrosis mosaic virus RNA 1 | NC_028144.1 |
| rice necrosis mosaic virus RNA 2 | NC_028145.1 |
| rose yellow mosaic virus | NC_019031.1 |
| ryegrass mosaic virus | NC_001814.1 |
| scallion mosaic virus | NC_003399.1 |
| shallot yellow stripe virus | NC_007433.1 |
| sorghum mosaic virus | NC_004035.1 |
| soybean mosaic virus | NC_002634.1 |
| squash vein yellowing virus | NC_010521.1 |
| sugarcane mosaic virus | NC_003398.1 |
| sugarcane streak mosaic virus | NC_014037.1 |
| sunflower chlorotic mottle virus | NC_014038.1 |
| sunflower mild mosaic virus isolate Entre Rios | NC_021065.1 |
| sunflower ring blotch virus isolate Chaco | NC_034208.1 |
| sweet potato feathery mottle virus | NC_001841.1 |
| sweet potato latent virus | NC_020896.1 |
| sweet potato mild mottle virus | NC_003797.1 |
| sweet potato virus 2 | NC_017970.1 |
| sweet potato virus C | NC_014742.1 |
| sweet potato virus G isolate Jesus Maria | NC_018093.1 |
| tall oatgrass mosaic virus isolate Benesov | NC_022745.1 |
| tamarillo leaf malformation virus isolate A | NC_026615.1 |
| telosma mosaic virus | NC_009742.1 |
| thunberg fritillary virus | NC_007180.1 |
| tobacco etch virus | NC_001555.1 |
| tobacco mosqueado virus isolate RS-01 | NC_030118.1 |

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
