# Peer review of "Peppers under Siege: Revealing the Prevalence of Viruses and Discovery of a Novel Potyvirus Species in Venezuela"

_sustainability, doi:10.3390/su152014825_

Round 1

Reviewer 1 Report

Dear Authors,

The study is a local survey study. The new virus you found (pepper severe mottle) has not been cross-validated. It cannot be said that it is a new virus with only sequence data. The sequence analysis should be compared with different virus (e.g. pepper mottle virus; pepper mild mottle virus... etc.) sequences. In fact, reinoculations should be made and sequence analysis should be done again. Thus, it is also revealed whether the virus claimed to be new is a strain of another virus. Although it is a virus study, the virus names were not written according to the latest spelling rules of the VIDE database. The virus writings were indited differently in different sections, which reduced the desire for an in-depth review and proofreading of the article for improvement.

In conclusion, although pepper viruses are a very important problem in cultivation areas, the determination of the occurrence of viruses is inadequate internationally. The findings of the study concern the region, it is not global. So, it is proposed that the manuscript should be evaluated for publication in local or regional journals.

 I wish you a successful science life.

-

Author Response

Response to Reviewer 1 Comments

Comments 1. The study is a local survey study. The new virus you found (pepper severe mottle) has not been cross-validated. It cannot be said that it is a new virus with only sequence data. The sequence analysis should be compared with different virus (e.g. pepper mottle virus; pepper mild mottle virus... etc.) sequences. In fact, reinoculations should be made and sequence analysis should be done again. Thus, it is also revealed whether the virus claimed to be new is a strain of another virus.

Response 1: Thank you for pointing this out. Data on the inoculation and reinoculation experiments, RT-PCR, and Sanger sequencing using the new virus as inoculum source was added (Fig. 5), cross-validating the new virus species. Table A2 showing the CP aa sequence comparison with different viruses was added.

Although it is a virus study, the virus names were not written according to the latest spelling rules of the VIDE database. The virus writings were indited differently in different sections, which reduced the desire for an in-depth review and proofreading of the article for improvement.

Response 2:  The Virus names were written according to ICTV code http://ictv.global/code

In conclusion, although pepper viruses are a very important problem in cultivation areas, the determination of the occurrence of viruses is inadequate internationally. The findings of the study concern the region, it is not global. So, it is proposed that the manuscript should be evaluated for publication in local or regional journals.

Response 3: Thank you for pointing this out. Although this is a local survey, results are relevant due to the international and often illegal exchange of plants, fruits, and seeds across the Colombian and Brazilian borderlands. We firmly believe that the identification of plant viruses could influence the implementation of certification schemes in Venezuela and neighboring countries, allowing the production of planting material of known variety and plant health status for local growers by controlling the propagation of pathogen-tested mother plants. This fact gains particular importance in a country like Venezuela, where plant disease outbreaks are causing substantial declines in major staple food and cash crops, impacting rural livelihoods and posing a significant and growing threat to the already complex food insecurity crisis (Marys and Rosales, 2021 https://doi.org/10.3389/fsufs.2021.715463). A recent literature review demonstrated that very few viral emergencies or novel threats have been mentioned in pepper crops in Venezuela during the past 20 years. New virus species found during this investigation were cross-validated as you’ll find below. For these reasons, we believe our findings contribute to global knowledge and are worth being published in the Sustainability special issue focused on Sustainable Crop Plants Protection.

  1.  

Reviewer 2 Report

During pepper production seasons in 2008 and 2014 in Venezuela's Lara and Miranda states, 271 plants were tested for virus infections using reverse transcription-PCR. The study revealed the prevalence of several viruses, with significant changes in viral community composition between the two years and identified pepper as a natural host for AMV and TRV in Venezuela. Additionally, a potential new virus species, proposed as Pepper severe mottle virus (PepSMoV), was detected. Strengths of the study include comprehensive virus detection, temporal comparisons, and practical agricultural implications. Weaknesses encompass the dated nature of the data being reported in 2023 and the lack of clarity about sampling methods.

 1.      Based on the dates mentioned in the abstract, the study focuses on data from 2008 and 2014, which would be quite dated if reported in 2023. This is a limitation.

 2.      The study is being reported or published in 2023, nearly a decade after the last set of data from 2014. This lag in publication can raise questions about the current relevance and applicability of the findings. The viral landscape, agricultural practices, and the environment can change significantly in a decade. Therefore, the data might not reflect the current situation of pepper production in Venezuela or elsewhere.

 3.      This time gap also raises questions about why there was such a delay in publication.

 4.      The study otherwise performed well and well discussed.

 5.      It will be better to include latest data, good quality images of viruses (each virus) and typical symptoms.

minor corrections required.

Author Response

Response to Reviewer 2 Comments

During pepper production seasons in 2008 and 2014 in Venezuela's Lara and Miranda states, 271 plants were tested for virus infections using reverse transcription-PCR. The study revealed the prevalence of several viruses, with significant changes in viral community composition between the two years and identified pepper as a natural host for AMV and TRV in Venezuela. Additionally, a potential new virus species, proposed as Pepper severe mottle virus (PepSMoV), was detected. Strengths of the study include comprehensive virus detection, temporal comparisons, and practical agricultural implications. Weaknesses encompass the dated nature of the data being reported in 2023 and the lack of clarity about sampling methods.

Response: Thank you for pointing this out. Line 93-98: The 2022 data was updated and the sampling methods were clarified

  1. Based on the dates mentioned in the abstract, the study focuses on data from 2008 and 2014, which would be quite dated if reported in 2023. This is a limitation.

  1. The study is being reported or published in 2023, nearly a decade after the last set of data from 2014. This lag in publication can raise questions about the current relevance and applicability of the findings. The viral landscape, agricultural practices, and the environment can change significantly in a decade. Therefore, the data might not reflect the current situation of pepper production in Venezuela or elsewhere.

  1. This time gap also raises questions about why there was such a delay in publication.

Response 1, 2 and 3: Regarding the dated nature of the results, we strongly suggest the read of the article “Plant Disease Diagnostic Capabilities in Venezuela: Implications for Food Security” ( https://doi.org/10.3389/fsufs.2021.715463), in which we detailed how the ongoing humanitarian, social and political crisis impacted on the phytosanitary services and hence the control of plant pests and diseases in the country.

We have only recently had international and national financing to complete the project on the pepper samples. Despite this fact, we believe our findings contribute to global knowledge and are worth being published in the Sustainability special issue focused on “Sustainable Crop Plants Protection”. We have now included data from a survey we made last year (2022) in one of the pepper production hotspots in the country (Miranda state). The results allow us to compare the viral species composition in the zone in 2014 and in 2022. Typical symptoms found in each zone are now reported.

  1. The study otherwise performed well and was well discussed.

Response 4: Thank you for pointing this out.

  1. It will be better to include latest data, good quality images of viruses (each virus) and typical symptoms.

Response 5: we agree. Revised and changed

Reviewer 3 Report

Manuscript ID: sustainability-2608886

Manuscript Title: Peppers Under Siege: Revealing the Prevalence of Viruses and Discovery of a Novel Potyvirus Species in Venezuela.

This study aimed to enhance the understanding of the complex interactions between viruses and pepper plants, paving the way for effective management strategies and safeguarding the future of pepper cultivation.

-Comments and Suggestions for Authors

The manuscript is clear, technically correct and well written. The English is overall ok, I have listed below some comments need to be corrected

-          In line 123, Oligo (dT) Primer is single-stranded sequence of deoxythymine (dT). Please correct Oligo d (T) to "Oligo (dT)".

-          In line 137, Please consider the note add under figure 4.

-          In line 176, remove the comma after diagnosis. The comma maybe separating the subject and verb in your sentence.

-          In line 274, lease complete the following note and add it to the figure legend "the Bootstrap values were calculated from 1000 replications and only the values with ......???% bootstrapping were considered significant, and are indicated on the branch nodes."

-          In conclusions section, please remind readers of your main points, remind readers of your evidence or arguments and wrap everything up by tying it all together.

-          In Table A1, Please mention in the table note the meaning of letters other than the four known letters; A, G, C, T. For example R = A or G, Y = C or T, S = G or C, W = A or T, K = G or T, M = A or C, B = C or G or T, D = A or G or T, H = A or C or T, V = A or C or G, N = any base.

-          In references section, please unify the style according to the journal instructions

(You can find the corrections in the annotated pdf)

Author Response

Response to Reviewer 3 Comments

This study aimed to enhance the understanding of the complex interactions between viruses and pepper plants, paving the way for effective management strategies and safeguarding the future of pepper cultivation.

The manuscript is clear, technically correct and well written. The English is overall ok, I have listed below some comments need to be corrected

-          In line 123, Oligo (dT) Primer is single-stranded sequence of deoxythymine (dT). Please correct Oligo d (T) to "Oligo (dT)".

Response 1. Oligo d (T) was corrected to "Oligo (dT)".

-          In line 137, Please consider the note add under figure 4.

Response 2. In line 137, the note added under Figure 4 was considered.

-          In line 176, remove the comma after diagnosis. The comma maybe separating the subject and verb in your sentence.

Response 3. In line 176, the comma after “diagnosis” was removed.

-          In line 274, please complete the following note and add it to the figure legend "the Bootstrap values were calculated from 1000 replications and only the values with ......???% bootstrapping were considered significant, and are indicated on the branch nodes."

Response 4. Bootstrap parameters are now completed. Line 342-344. Bootstrap values (n = 1,000) or probability estimate values larger than 70% are indicated at branch nodes for neighbor-joining/maximum-likelihood/Bayesian analysis. Each colored box indicates major phylogenetic groups

-          In conclusions section, please remind readers of your main points, remind readers of your evidence or arguments and wrap everything up by tying it all together.

Response 5. Suggestions were taken into account for the Conclusions section.

-          In Table A1, Please mention in the table note the meaning of letters other than the four known letters; A, G, C, T. For example R = A or G, Y = C or T, S = G or C, W = A or T, K = G or T, M = A or C, B = C or G or T, D = A or G or T, H = A or C or T, V = A or C or G, N = any base.

Response 6. In Table A1, IUPAC nucleotide code was added for letters other than A, G, C, T

-          In references section, please unify the style according to the journal instructions

Response 7. References were unified accordingly.

Reviewer 4 Report

I have reviewed the manuscript, "Peppers Under Siege: Revealing the Prevalence of Viruses and Discovery of a Novel Potyvirus Species in Venezuela,” submitted for publication in Sustainability. The provided information appears to summarize a scientific study or report on virus outbreaks in pepper plants in Venezuela during 2008 and 2014. I appreciate the effort put into the study. However, after careful consideration, I believe the manuscript requires significant revisions before being considered for publication. Here are some critical comments and observations on the information provided:

#The abstract provides information about virus outbreaks in pepper plants in Venezuela but lacks context regarding the broader agricultural and environmental factors that might have contributed to these outbreaks. Understanding the underlying causes and conditions is important for developing effective mitigation strategies.

Refine the abstract to reflect the key findings and contributions of the study accurately.

# The manuscript needs a clearer statement of the research objectives and hypotheses. It should explicitly state what questions the study aims to answer and what specific goals it seeks to achieve.

# The methods section lacks detail regarding the techniques and procedures used during the surveys and in characterizing the potyvirus. It is crucial to provide a step-by-step description of the methods used, including sample size, sampling locations, and laboratory techniques.

#Mention the source of reference sequences for phylogenetic analysis to ensure reproducibility.

# The summary presents data on the prevalence of different viruses in pepper plants during the two survey years (2008 and 2014). However, it does not provide details about the sample size or the specific methods used for the surveys, which are important for assessing the robustness of the findings.

# Identifying various viruses in pepper plants is significant for understanding the health of these crops. However, it would be helpful to provide more details about the techniques and criteria used for virus identification to ensure the accuracy of the results.

# The summary mentions the proposal of a new virus species, Pepper severe mottle virus (PepSMoV), based on the complete capsid protein (CP) sequence. It would be important to provide more information about the taxonomic criteria and the process used to establish this new species, as this has implications for virus classification and naming within the scientific community.

#  The summary discusses the value of knowledge about the prevalence and distribution of viral species for recommending virus-resistant cultivars. However, it would be beneficial to include information on the economic and agricultural impact of these virus outbreaks to highlight the importance of the research.

The discussion section should delve into the implications of the findings in greater detail. Address the significance of the results for pepper cultivation in Venezuela, potential risks to other crops, and the broader implications for virus management strategies.

Explore the reasons behind the observed changes in viral community composition between 2008 and 2014.

# Revise the conclusions to ensure they align with the objectives and findings of the study. Provide clear, actionable recommendations based on the results. Reorganize the manuscript to improve its overall flow and coherence. Ensure that each section logically follows the previous one.

#The summary provides descriptive information about the virus outbreaks and their prevalence but does not delve into a detailed discussion of the implications of these findings. It would be helpful to discuss the potential consequences for pepper production in Venezuela, the strategies for managing these outbreaks, and the importance of ongoing research in this area.

# Since the knowledge cutoff date is September 2021, it would be important to include any updates or developments in research related to these virus outbreaks if available. Research in this field may have progressed, and new findings could provide additional insights.

# In conclusion, while the provided information highlights the prevalence of different viruses in pepper plants in Venezuela, there is a need for more contextual information, detailed methodology, and discussion of the implications of the findings to provide a comprehensive understanding of the situation and its significance for agriculture in the region.

Carefully proofread the manuscript for grammar, punctuation, and language errors.

Minor editing of English language required.

Author Response

Response to Reviewer 4 Comments

I have reviewed the manuscript, "Peppers Under Siege: Revealing the Prevalence of Viruses and Discovery of a Novel Potyvirus Species in Venezuela,” submitted for publication in Sustainability. The provided information appears to summarize a scientific study or report on virus outbreaks in pepper plants in Venezuela during 2008 and 2014. I appreciate the effort put into the study. However, after careful consideration, I believe the manuscript requires significant revisions before being considered for publication. Here are some critical comments and observations on the information provided:

#The abstract provides information about virus outbreaks in pepper plants in Venezuela but lacks context regarding the broader agricultural and environmental factors that might have contributed to these outbreaks. Understanding the underlying causes and conditions is important for developing effective mitigation strategies.

Refine the abstract to reflect the key findings and contributions of the study accurately.

Response 1. Thank you for pointing this out. The abstract section was refined to reflect key findings and contributions. Also, factors that may have contributed to virus outbreaks are pointed out.

# The manuscript needs a clearer statement of the research objectives and hypotheses. It should explicitly state what questions the study aims to answer and what specific goals it seeks to achieve.

Response 2. Research objectives and hypotheses are now stated in the Introduction section. Hypothesis: In the pepper crops of Venezuela, there is a significant prevalence of known viral pathogens, and it is possible that novel virus species, particularly within the Potyvirus genus, may also exist. Understanding the extent of viral prevalence and identifying potential novel viral species is essential for effective disease management and sustainable pepper cultivation in the region."

# The methods section lacks detail regarding the techniques and procedures used during the surveys and in characterizing the potyvirus. It is crucial to provide a step-by-step description of the methods used, including sample size, sampling locations, and laboratory techniques.

Response 3. Procedures used during surveys and virus characterization are pointed out, including sample size, sampling locations, and laboratory techniques.

#Mention the source of reference sequences for phylogenetic analysis to ensure reproducibility.

Response 4. The source of reference sequences for phylogenetic analysis is presented in Table A2.

# The summary presents data on the prevalence of different viruses in pepper plants during the two survey years (2008 and 2014). However, it does not provide details about the sample size or the specific methods used for the surveys, which are important for assessing the robustness of the findings.

Response 5. Line 19-29.  Details about techniques and criteria used for virus identification (RT-PCR amplification with primers for virus groups or specific for the putative novel virus, cloning and sequence analysis of amplicons) were added.

# Identifying various viruses in pepper plants is significant for understanding the health of these crops. However, it would be helpful to provide more details about the techniques and criteria used for virus identification to ensure the accuracy of the results.

# The summary mentions the proposal of a new virus species, Pepper severe mottle virus (PepSMoV), based on the complete capsid protein (CP) sequence. It would be important to provide more information about the taxonomic criteria and the process used to establish this new species, as this has implications for virus classification and naming within the scientific community.

Response 6 and 7. Information about the taxonomic criteria and the process used to establish putative new species is stated.

#  The summary discusses the value of knowledge about the prevalence and distribution of viral species for recommending virus-resistant cultivars. However, it would be beneficial to include information on the economic and agricultural impact of these virus outbreaks to highlight the importance of the research.

The discussion section should delve into the implications of the findings in greater detail. Address the significance of the results for pepper cultivation in Venezuela, potential risks to other crops, and the broader implications for virus management strategies.

Response 8. The potential agricultural and economic impact of virus outbreaks in pepper highlighting the importance of the study is pointed out. The reasons behind changes in the viral community composition are now explicit.

Explore the reasons behind the observed changes in viral community composition between 2008 and 2014.

Response 9. Line 435-451. The reasons behind changes in the viral community composition are now explicit.

# Revise the conclusions to ensure they align with the objectives and findings of the study. Provide clear, actionable recommendations based on the results. Reorganize the manuscript to improve its overall flow and coherence. Ensure that each section logically follows the previous one.

Response 10. Conclusions are now aligned with the objectives and findings.

#The summary provides descriptive information about the virus outbreaks and their prevalence but does not delve into a detailed discussion of the implications of these findings. It would be helpful to discuss the potential consequences for pepper production in Venezuela, the strategies for managing these outbreaks, and the importance of ongoing research in this area.

Response 11. Done. 32-26

# Since the knowledge cutoff date is September 2021, it would be important to include any updates or developments in research related to these virus outbreaks if available. Research in this field may have progressed, and new findings could provide additional insights.

Response 12. Done. The 2022 data was updated

# In conclusion, while the provided information highlights the prevalence of different viruses in pepper plants in Venezuela, there is a need for more contextual information, detailed methodology, and discussion of the implications of the findings to provide a comprehensive understanding of the situation and its significance for agriculture in the region.

 Response 13. Thank you for pointing this out. The changes are shown in the corrected version of the manuscript

Carefully proofread the manuscript for grammar, punctuation, and language errors.

Response 14. Done.

Round 2

Reviewer 1 Report

Dear Authors,

When I reviewed the article for the second time, it was seen that you mostly took into account my and other reviewers' opinions and suggestions.

However, it has been observed that you did not take into account some of my opinions in the attached file. When your views are contrary, it is expected to make clarification of why you are contrary.

I wish a successful science life.

Reviewer 2 Report

The manuscript can now be accepted as it is revised thoroughly. 

Reviewer 4 Report

The authors have revised the manuscript as per the suggested lines and now it can be accepted in its current form.